# Chloroplast Damage and Photosynthetic System Disorder Induced Chlorosis in the Leaves of Rice Seedlings under Excessive Biuret

Peng Zhang [1,2], Yulin Chen [1], Yuping Zhang [1], Jing Xiang [1], Yaliang Wang [1], Zhigang Wang [1], Huizhe Chen [1,*] and Yikai Zhang [1,*]

[1] State Key Laboratory of Rice Biology and Breeding, China National Rice Research Institute, Hangzhou 310006, China; pengz199606@163.com (P.Z.); wangyaliang@caas.cn (Y.W.)

[2] College of Landscape and Ecological Engineering, Hebei University of Engineering, Handan 056000, China

[*] Correspondence: chenhuizhe@caas.cn (H.C.); zhangyikai@caas.cn (Y.Z.)

**Abstract:** Excessive biuret in fertilizer causes leaf albinism in direct-seeded rice fields. This study aimed to provide a comprehensive understanding of the underlying physiology and molecular mechanisms of leaf chlorosis via biuret using morphophysiological and transcriptome analyses. The induction of biuret in albino rice leaves was examined in a net-growing cultivation bed. Some key morphophysiological indices were measured including biuret content, blade ultrastructure, chlorophyll content, and chlorophyll fluorescence parameters. Candidate genes in the chlorotic leaves under biuret stress were also excavated using transcriptome analysis. Furthermore, physiological and biochemical analyses of the changes in enzyme activities and intermediate metabolite contents in relation to the phenotypic changes in the leaves were carried out. The chlorotic leaves of rice seedlings showed higher biuret accumulation, and the leaves suffered severe damage with higher malondialdehyde contents and low chlorophyll contents. Abnormal chloroplast ultrastructures and thylakoid membrane structure loss were observed in chlorotic leaves under biuret exposure. The related genes involved in the chloroplast development, photosynthesis (including antenna proteins), and carbon fixation pathways were significantly downregulated, which suggests that photosynthesis was destroyed in the chlorotic leaves of rice seedlings. Biuret disturbed the photosynthetic system in chloroplast thylakoid membranes by inhibiting chloroplast development, thereby promoting the formation of the chlorotic leaf phenotype in rice seedlings. Our results promote the understanding of the molecular mechanism of rice in response to biuret toxicity.

**Keywords:** rice; biuret; chloroplasts; photosynthetic systems; morphophysiological and transcriptome analyses

## 1. Introduction

Rice (*Oryza sativa* L.) is one of the most influential grain crops in China. The planting area and total yield of rice occupy an important position in grain production in China, which is of great significance to ensure food security [1]. In recent years, with the gradual transfer of the rural labor force to secondary and tertiary industries and the increase in labor costs, the rice planting mode has changed from single manual transplanting to mechanized transplanting and direct seeding, saving labor and costs [2]. Compound fertilizers are commonly used in the direct seeding of rice because they provide more main nutrients for rice growth. In compound fertilizer production, urea has become the preferred nitrogen source due to its high nitrogen content [3]. Biuret is the product of urea decomposition. Excessively high temperatures in the process of granulation and drying result in the condensation and deamination of two urea molecules to form biuret [4,5]. Highly concentrated biuret in fertilizer or pure biuret could have a toxic effect on crops when applied to the soil or via foliage spraying [6].

The standard for the permissible concentration of biuret is lower than 1.2% for urea fertilizers [6]. However, there is no corresponding national standard for biuret content in compound fertilizer. After applying urea or compound fertilizer containing biuret exceeding this standard, farmers may cause harm via burning seedlings, thus leading to huge economic losses and harming the vital interests of farmers [6]. Recently, there have been many occurrences of the chlorotic leaf phenomenon in direct-seeded rice fields, mainly due to excessive biuret in fertilizers [7].

Chlorophyll is the primary factor involved in leaf color formation and is produced in the chloroplasts. The composition of chloroplasts includes the chloroplast membrane, thylakoids, and stroma, and structural changes in the chloroplast directly result in leaves losing their green color [8]. High concentrations of biuret can cause an imbalance in plant nutrients, especially nitrogen-source compounds, and induce the accumulation of excessive reactive oxygen species (ROS) in plant tissues, causing typical oxidative stress in plants [9]. As a result, the structure and function of plant cells are destroyed, and the chlorophyll content in the leaves is decreased due to the adverse environment [7,10]. External environmental influences such as light, temperature, water, and mineral elements account for albinism in plant leaves; however, apart from these external factors, leaf albinism in plants is inherently and primarily controlled by internal genes.

Plant leaf albinism is determined by the expression levels of internal genes, such as genes involved in chloroplast development, chloroplast division, and chlorophyll biosynthesis [11]. The rice leaf color phenotype is mostly controlled by a pair of recessive nuclear genes, which are among more than 150 genes related to chlorophyll content contained in the Gramene database (http://www.gramene.org/ (accessed on 15 February 2023)). These genes directly or indirectly regulate chlorophyll metabolism, photosynthesis, carbon fixation, energy metabolism, ribosome metabolism, post-translational modification, and anthocyanin biosynthesis, which affect leaf color changes in a variety of ways [12–14].

Biuret toxicity to crops is often reported in fertilization; however, the mechanism through which biuret causes albinism in rice leaves remains unclear. In the present study, morphophysiological and transcriptome analyses were conducted to reveal the physiological and molecular mechanisms of chlorosis in the leaves of seedlings under biuret stress. The results could promote the understanding of the molecular mechanism of rice in response to biuret toxicity.

## 2. Materials and Methods

### 2.1. Plant Material and Growth Conditions

The test soil type was paddy soil, and the experiment was conducted in a net-room cultivation bed (4 m × 1 m × 0.5 m, length × width × height) at the experimental base of the China Rice Research Institute in Fuyang, China. The acidified paddy soil (pH 5.53) contained 1.64 g kg$^{-1}$ of total N, 18.92 mg kg$^{-1}$ of available P (Olsen-P), 50.02 mg kg$^{-1}$ of available K, and 40.93 g kg$^{-1}$ of organic matter. Rice seeds (*Oryza sativa* L., cv. Yongyou1540) were purchased from Zhejiang Ningbo Seed Industry Co., Ltd. The seeds were soaked for 48 h and directly seeded in the cultivation bed at a planting density of 80 g per bed. The experiment was conducted at a daytime temperature of 30–35 °C and a nighttime temperature of 22–26 °C under natural light conditions.

During the experiment, biuret was added at rates of 0, 30, and 60 mg biuret kg$^{-1}$ of soil. The amounts of soil nutrients added were as follows (in mg kg$^{-1}$ soil): 200 of N (as $(NH_4)_2SO_4$), 120 of P (as superphosphate), and 150 of K (as KCl). Initially, the biuret required for the experiment was weighed and subsequently blended evenly with the fertilizer. Prior to sowing, the seedbed was irrigated, plowed, and leveled, and then the fertilizer mixture was uniformly distributed onto the seedbed and incorporated into the soil to a depth of 10 cm and sowed. All the experiments were performed with three bio-replicates. According to the rice seedling growth and physiological experimental data, plants from the control and the biuret treatments were sampled at 3 and 7 days after seedling emergence, respectively. Subsequently, seedlings were sampled, and physiological indexes

such as plant height, root length, and shoot weight were determined. Gene expression was also analyzed using high-throughput sequencing.

### 2.2. Biuret Concentration in Rice Seedling Assays

During sampling, seedlings were separated into shoots and roots, which were rinsed with deionized water. After the fresh weights were determined, the rice seedlings were freeze-dried for 2 days. The crushed plant samples were weighed and placed in a centrifuge tube (50 mL). Anhydrous ethanol was added, and the samples were washed in an ultrasonic cleaner for 60 min. Then, the samples were centrifugated at 8000 rpm for 10 min at a temperature of 25 °C, and the supernatants were transferred into an evaporating dish and dried in a water bath. After adding ultrapure water, the residues were shaken and dissolved ultrasonically. The impurities were removed with a needle filter membrane (0.45 μm) for HPLC analysis.

A 20 μL supernatant was injected into an HPLC system (LC-10AS; UV detector SPD-10A, Shimadzu, Kyoto, Japan) equipped with a hydrophilic interaction chromatography column (Venusil XBP–C8, Agela Technologies Inc., Wilmington, CA, USA). The mobile phase comprised a water phase buffer solution ($\varrho$ ($KH_2PO_4$) 1 g/100 mL potassium dihydrogen phosphate solution; adjusted pH = 6.0 with phosphoric acid solution) and methanol (95:5 volume ratio). The flow rate was 1 mL min$^{-1}$, the detection wavelength was 200 nm, and the retention time of the biuret was approximately 3.8 min.

### 2.3. Chlorophyll Abundance and Chloroplast Ultrastructure Observation

After one week of seedling emergence, rice plants with uniform growth were selected. A total of 0.1 g of fresh leaves was taken, cut into fragments, and extracted with 5 mL of 95% alcohol for 48 h in the dark until the leaves became white. The light absorption value of the leaves was measured at 665 nm, 649 nm, and 470 nm using a SPECORD 200 spectrophotometer (AnalytikJena, Jena, Germany) according to a previous publication [15].

The sampled rice leaves were cut into 1 cm$^2$ pieces with a blade and fixed in a 2.5% glutaraldehyde solution for 24 h at 4 °C. After rinsing with 0.1 M phosphate buffer (pH 7.0), the sample was fixed in 1% osmium tetroxide ($OSO_4$) for 2 h and rinsed with 0.1 M phosphate buffer (pH 7.0) for 15 min. The sample was dehydrated with ethanol gradient concentrations of 30%, 50%, 70%, 80%, 90%, and 100%. Subsequently, the dehydrated sample was sliced using an ultramicrotome (Leica Microsystems, Ltd., Wetzlar, Germany), and 70–90 nm sections were obtained. The sections were stained with lead citrate solution and a 50% ethanol-saturated solution of hydrogen peroxide acetate for 5–10 min, and the ultrastructure of the leaf cells was observed using a transmission electron microscope (HT-7650, Hitachi, Tokyo, Japan).

### 2.4. Chlorophyll Fluorescence Assays

For the treatment group, the first leaves were selected for measurement. The live fluorescence of the rice leaves was measured using a multi-function modulated fluorescence imaging system IMAGING-PAM (Walz, Bavaria, Germany). Before measuring, the sample was put in the dark for 30 min, and then the initial fluorescence ($F_o$), maximum fluorescence ($F_m$), non-photochemical quenching (NPQ, $q^N$), and photochemical quenching ($q^P$) were measured [16]. The maximal photochemical efficiency ($F_v/F_m = (F_m - F_o)/F_m$) was calculated according to a previous study [17]. The effective PSII quantum yield ($\Phi_{PSII}$) was determined as ($F_m' - F_s)/F_m'$ [18]. The photosynthetic electron transport rate (ETR) was determined as ETR = PAR $\times \Phi_{PSII} \times 0.84 \times 0.5$ [19].

### 2.5. RNA Extraction, Sequencing, and Mapping

Total RNA was extracted from the leaf tissues after exposure to biuret for 3 and 7 days using an RNeasy® Mini kit (Qiagen, Hilden, Germany). Three bio-replicates were performed for each treatment. The quality of the RNA was detected using 1% agarose

gel, and the concentration and integrity were detected using the Bioanalyzer 2100 system (Agilent Technologies, Santa Clara, CA, USA). After qualified detection, magnetic beads with Oligo (dT) were used to enrich the eukaryotic mRNA, and Fragmentation Buffer was used to randomize the interruption of mRNA. Using RNA as a template, a six-base random primer and M-MuLV Reverse Transcriptase (RNase H-) were used to synthesize the first-strand cDNA. Then, the second cDNA chain was synthesized by adding buffer, dNTPs, RNase H, and DNA polymerase I with cDNA purified with AMPure XP beads. The purified double-stranded cDNA was then repaired, A-tailed, and ligated with a sequencing adapter. The fragment size was selected using AMPure XP beads, and finally, the cDNA library was obtained via PCR enrichment. The library was sequenced using the Illumina NovaSeq 6000 platform (Illumina Inc., San Diego, CA, USA). Each sample produced an average of 6 billion clean data. Subsequently, raw reads containing connectors and low-quality reads were filtered, and high-quality reads were obtained for all downstream analyses. The reference genome and the gene model annotation files were compared with the Nipponbare reference genome (Oryza_sativa.IRGSP-1.0.dna.toplevel.fa).

*2.6. Differentially Expressed Gene (DEG) Analysis*

The gene expression levels were expressed in units of the fragment number per kilobase of transcript sequence per millions of base pairs sequenced (FPKM) using RSEM software (v1.2.15) [20]. DEGs were detected using DESeq version 1.8.2 and screened using the following parameters: expression difference multiple $|\log2\text{FoldChange}| > 1$ and adjusted $p$-value of $\leq 0.05$. Gene ontology (GO) enrichment was analyzed among DEGs based on $p$-values of less than 0.05 utilizing the GOseq R package version 3.5.0. In addition, a Kyoto Encyclopedia of Genes and Genomes (KEGG) pathway analysis was conducted using the KOBAS (2.0) software to determine the number of DEGs involved in metabolic pathways. Principal component analysis (PCA) and correlation diagrams were mainly produced using PRCOMP and SCATTERPLOT3D, and the heatmap was drawn using the "pheatmap" function in the R package (R4.1.1).

*2.7. Quantitative Real-Time PCR (qRT-PCR) Analysis*

Total RNA was extracted using an RNeasy® Mini kit (Qiagen, Hilden, Germany) in accordance with the RNeasy® Mini instruction booklet. A qRT-PCR analysis was performed to validate the DEGs expression with the SYBR® Premix Ex TaqTM Kit (TaKaRa Biotechnology Co., Ltd., Dalian, China) using the ABI Q3 Real-Time PCR System (Applied Biosystems, Grand Island, NE, USA). NM 197,297, an actin gene in rice, was utilized as a housekeeping gene. The PCR assays were undertaken in accordance with a previous report [21]. Each sample was subjected to 4 technical replicates, and the relative quantitative method ($2^{-\Delta\Delta CT}$) was employed to estimate the relative expression levels [22].

*2.8. Assay of Enzyme Activities and Membrane Permeability of Leaves*

Plant leaves were collected after exposure to biuret for 7 days and ground into a powder by adding a suitable amount of liquid nitrogen into a pre-cooled mortar. Ribulose-1,5-bisphosphate carboxylase (Rubisco) activity was determined as described in a previous report [23]. The activity of glyceralde-hyde-3-phosphatedehydrogenase (GAPDH) was assayed according to a previous publication [24]. The activities of D-ribulose-5-phosphate 3-epimerase (RPE), lipoxygenase activity (LOX), red chlorophyll catabolite reductase (RCCR), and ascorbate peroxidase (APX) were evaluated according to previous studies [25–27]. The degree of membrane lipid peroxidation in fresh leaves was evaluated based on the content of MDA and determined using the thiobarbituric acid method [28]. The membrane permeability of fresh leaves was determined based on the relative electrical conductivity (%) [29].

*2.9. Statistics*

The data were analyzed using one-way analyses of variance with a general linear model in SAS version 9.2 software (SAS Institute, Cary, NC, USA). The data are presented as means with the standard errors below. Statistically significant differences ($p < 0.05$) between averages were identified using *t*-tests.

## 3. Results

*3.1. Biuret Caused Leaf Albinism and Destroyed the Structure of Chloroplasts*

Figure 1 shows the results of the growth of rice seedlings treated with biuret at different concentrations on the 3rd and 7th days. The seedling leaves showed chlorosis after the seedlings emerged after 3 days. On the 3rd day, the plant heights, root lengths, and fresh weights of the shoots of rice seedlings treated with 30 and 60 mg kg$^{-1}$ of biuret were decreased by 7.7%, 0.3%, and 0.2%, and by 18.0%, 8.9%, and 5.1%, compared with the control, respectively. On the 7th day, the plant heights, root lengths, and fresh weights of the shoots of rice seedlings treated with 30 and 60 mg kg$^{-1}$ of biuret were decreased by 8.5%, 29.8%, and 1.0%, and by 27.7%, 39.1%, and 16.7%, compared with the control, respectively (Figure 1B(a–c)).

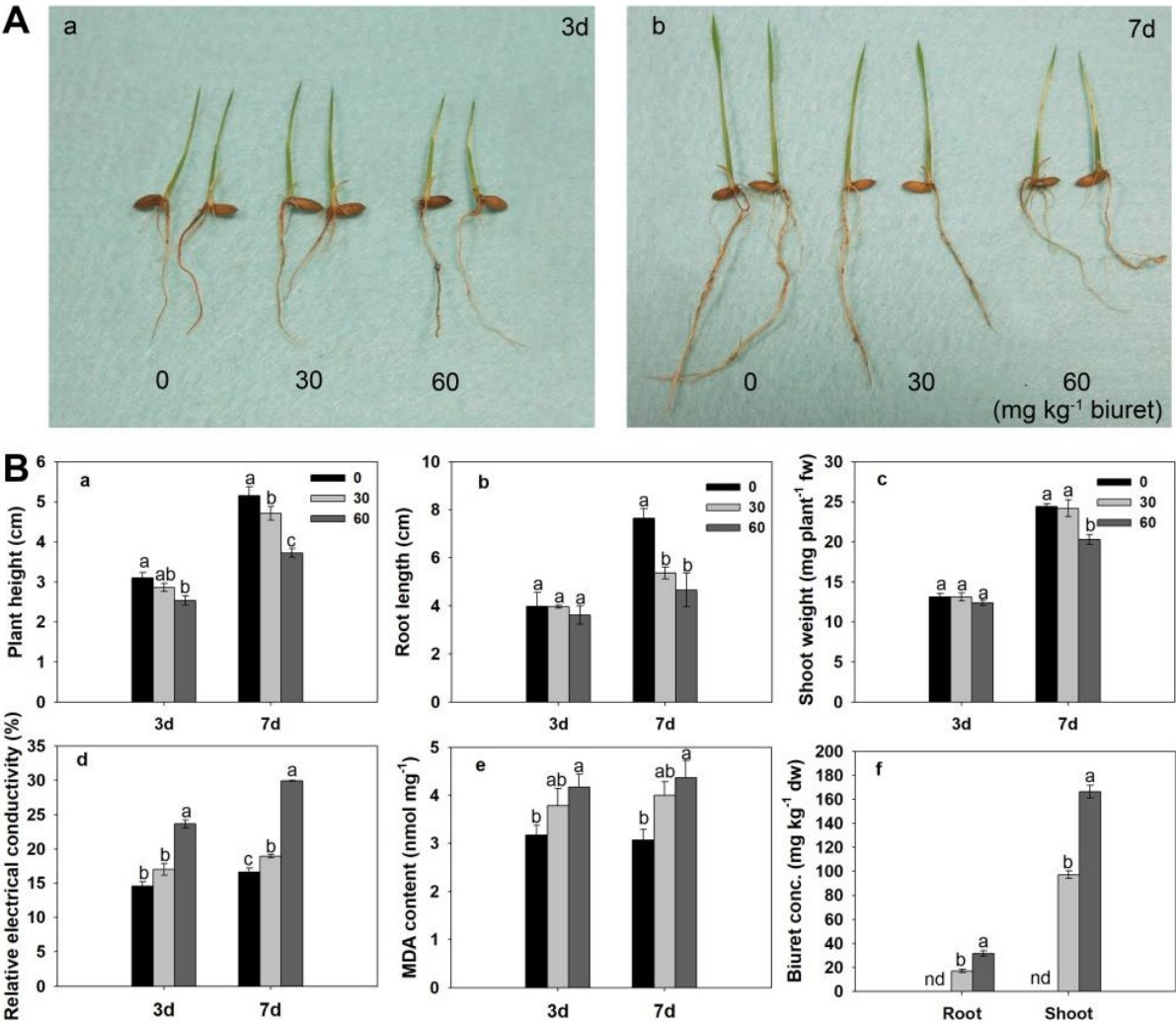

**Figure 1.** Effect of biuret toxicity on the growth of rice seedlings. (**A**) Phenotypic characteristics of rice seedlings under CK (a) and biuret treatment (b); (**B**) plant height (a), root length (b), shoot weight (c), relative electrical conductivity (d), MDA content (e), and biuret concentration (f) of rice seedling. Values are means ± SE (three independent biological replicates). Ten seedlings were combined for a representative sample. Different letters above the bars designate significant differences at $p < 0.05$.

Excessive ROS induced lipid peroxidation in stressed cells, and the MDA content and relative electrical conductivity were significantly increased by 30.1% and 41.9% and by 14.0% and 80.2% when exposed to 30 and 60 mg kg$^{-1}$ of biuret for 7 days, respectively (Figure 1B(d,e)). Biuret concentrations were determined in 7-day-old rice seedlings, which increased with rising biuret levels in the shoots and roots. However, no biuret was detected in the control, indicating that excessive biuret in plants is the cause of biuret damage (Figure 1B(f)).

To study the effect of biuret on the ultrastructure of rice chloroplasts, the leaf subcellular structure was observed via transmission electron microscopy. The leaves had typical cell and chloroplast structures in the control. Chloroplasts had osmium-containing granules, thylakoids, and basal granules. The morphology and structure of the cells and chloroplasts in chlorotic leaves treated with biuret changed significantly, with severely swollen chloroplasts, absent internal structures, and impaired chloroplast membranes (Figure 2), which indicated that the loss of chloroplast structure and thylakoid membrane structure under the biuret treatment was an important factor that caused the chlorotic phenotype.

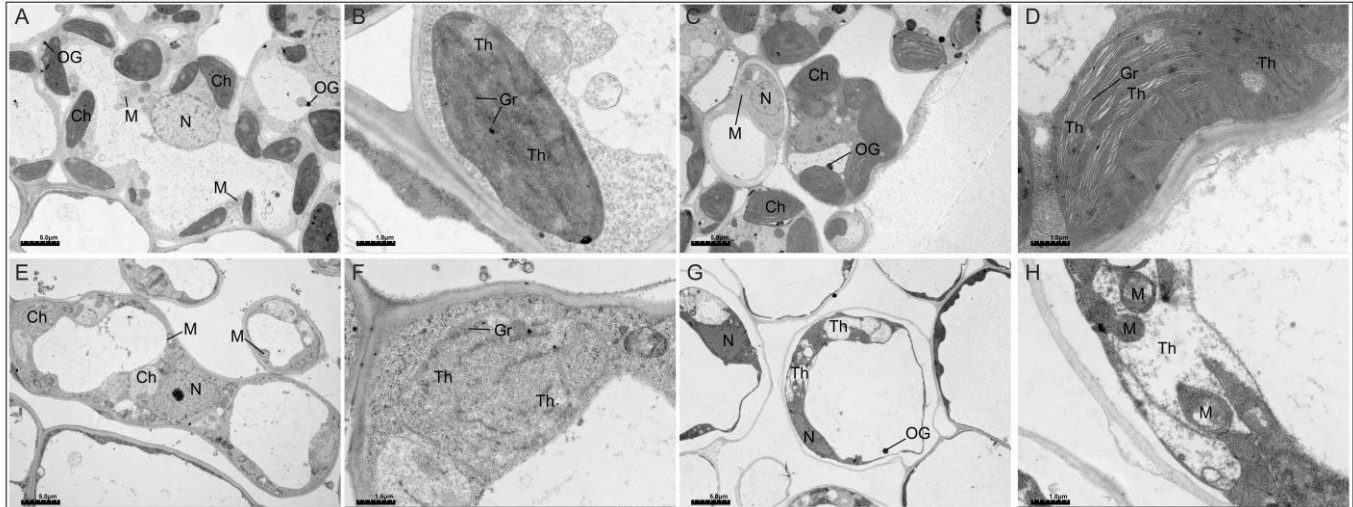

**Figure 2.** The chloroplast ultrastructures in rice seedling leaves exposed to biuret stress. (**A,B**) Chloroplast ultrastructure under non-biuret stress for 3 days, (**C,D**) chloroplast ultrastructure under non-biuret stress for 7 days, (**E,F**) chloroplast ultrastructures under biuret stress for 3 days, (**G,H**) chloroplast ultrastructures under biuret stress for 7 days. Bar = 5 μm in (**A,C,E,G**); bar = 1 μm in (**B,D,F,H**). Ch: chloroplast, M: mitochondria, Th: thylakoid grana, Gr: grana, OG: osmium granule.

*3.2. Biuret Decreased Leaf Pigment Content and Chlorophyll Fluorescence Parameters*

Biuret treatment significantly reduced the chlorophyll content in rice seedlings. When the rice seedlings were treated with 30 and 60 mg kg$^{-1}$ of biuret, the contents of Chl a, Chl b, total Chl, and carotenoids decreased by 6.7%, 5.9%, 6.5%, and 6.0%, and by 22.3%, 17.7%, 20.9%, and 21.0%, on the 3rd day, respectively. When the rice seedlings were treated with 30 and 60 mg kg$^{-1}$ of biuret, the contents of Chl a, Chl b, total Chl, and carotenoids significantly decreased by 17.3%, 13.2%, 16.2%, and 18.8%, and by 49.3%, 38.6%, 46.3%, and 48.1%, on the 7th day, respectively (Figure 3).

Chloroplasts are organelles used in photosynthesis in plants. To test whether the photosynthesis devices in chlorotic leaves were affected, we investigated some key parameters using a pulsed amplitude modulation chlorophyll fluorometer. Compared with the control, the $F_v/F_m$ significantly decreased by 21% and 29% with the application of 30 and 60 mg kg$^{-1}$ of biuret, respectively, while the $F_o$ increased by 23% and 32% with the application of 30 and 60 mg kg$^{-1}$ of biuret, respectively (Figure 4A,C). When the rice seedlings were treated with 30 and 60 mg kg$^{-1}$ of biuret, the $\Phi_{PSII}$ markedly decreased by 30% and 47%, respectively, the ETR decreased by 30% and 61%, respectively, the NPQ decreased by 6% and 16%, respectively, and the $q^P$ decreased by 10% and 29%, respectively,

while the $q^N$ obviously increased by 5% and 6%, respectively (Figure 4D–H). The results show that higher levels of biuret can significantly harm the PSII reaction center, inhibit the transfer of absorbed light energy to the reaction center, and reduce the conversion efficiency of the PSII reaction center.

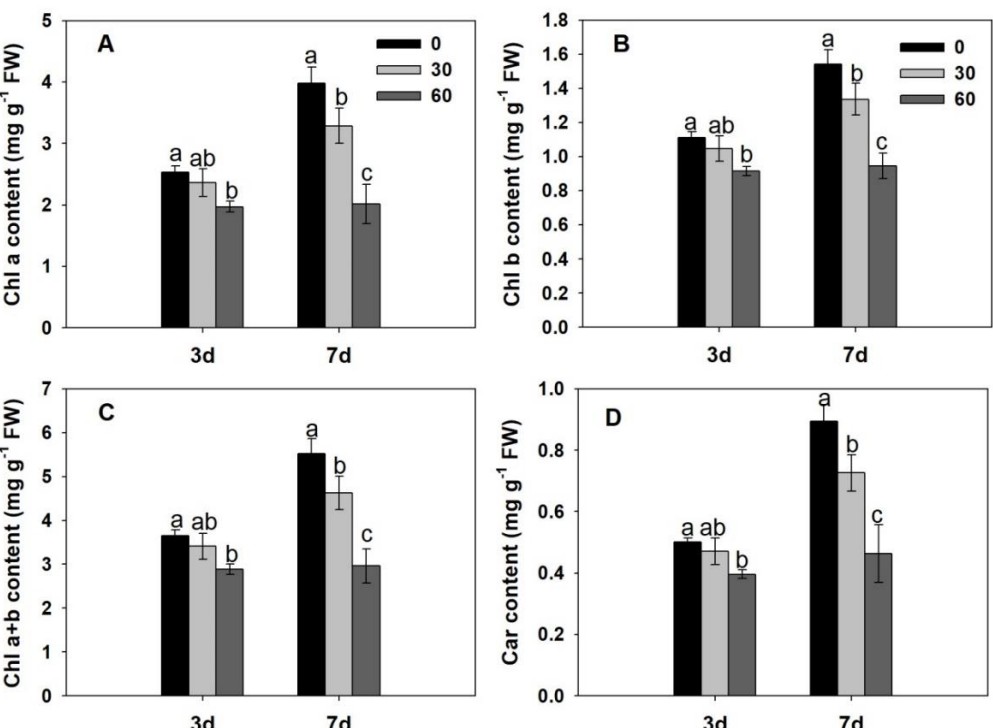

**Figure 3.** Chlorophyll abundance in rice seedling leaves exposed to biuret stress. (**A**) Chlorophyll a content in rice seedling leaves exposed to biuret stress. (**B**) Chlorophyll b content in rice seedling leaves exposed to biuret stress. (**C**) Chlorophyll a and b content in rice seedling leaves exposed to biuret stress. (**D**) Carotenoid content in rice seedling leaves exposed to biuret stress. Values are means $\pm$ SE (three independent biological replicates). Ten seedlings were combined for a representative sample. Different letters above the bars designate significant differences at $p < 0.05$.

### 3.3. Genes Involved in Chloroplast Biosynthesis Were Downregulated under Biuret Stress

To further investigate the molecular mechanism for the albinism caused by biuret application, transcriptome analysis was conducted using high-throughput sequencing. Twelve libraries were constructed from two time points (3 days and 7 days) after germination under biuret application. After trimming adapter sequences and low-quality reads, 44.1–50.7 million clean reads were finally produced for each sample (Table S1). More than 95.0% of the clean reads for each sample were mapped onto the reference genome (Table S1).

The first two principal components (PCs) from the PCA on the RNA sequencing data explained 94% of the total variability (Figure 5A). Germination days as the key factor had the strongest impact on the gene expression profiles, and samples from different germination days were separated by the first principal component (PC1) (Figure 5A). The treatment factor (PC2) had a lower impact on the number of genes, accounting for 5% of the variability; however, the samples taken 7 days after germination became more scattered than those taken at 3 days (Figure 5A), indicating that biuret had a greater effect on samples taken 7 days after germination. Moreover, pairwise comparison revealed a relatively small number of biuret-responsive DEGs at both 3 days and 7 days after germination, with 802 DEGs (569 upregulated and 233 downregulated) and 1694 DEGs (1157 upregulated and 537 downregulated) in T3d/CK3d and T7d/CK7d, respectively (Figure 4B). Noticeably, 9571 and 8745 DEGs were differentially expressed in CK7d/CK3d and T7d/T3d, respectively

(Figure 5B). In all four pairwise comparisons, all DEGs could be divided into 15 disjointed subgroups, and 79 DEGs belonged to common biuret-responsive genes (Figure 5C).

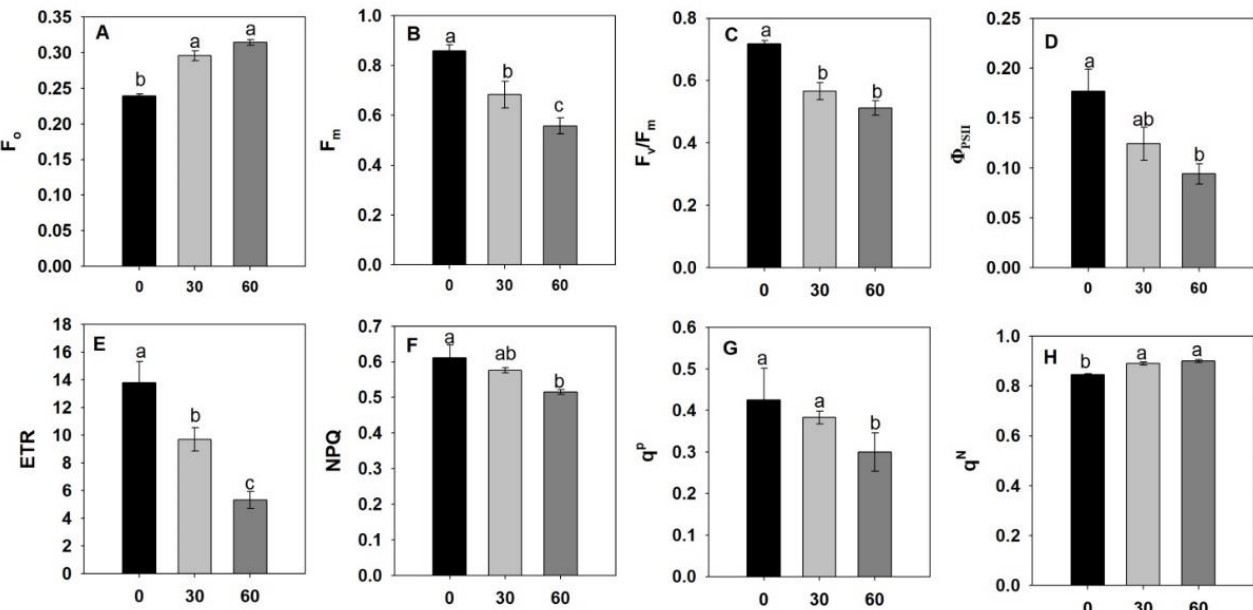

**Figure 4.** Changes in fluorescence parameters of rice seedlings exposed to biuret stress for 7 days $F_0$ (**A**), $F_m$ (**B**), $F_v/F_m$ (**C**), $\Phi_{PSII}$ (**D**), ETR (**E**), NPQ (**F**), $q^P$(**G**) and $q^N$ (**H**). Values are means ± SE (three independent biological replicates). Different letters above the bars designate significant differences at $p < 0.05$.

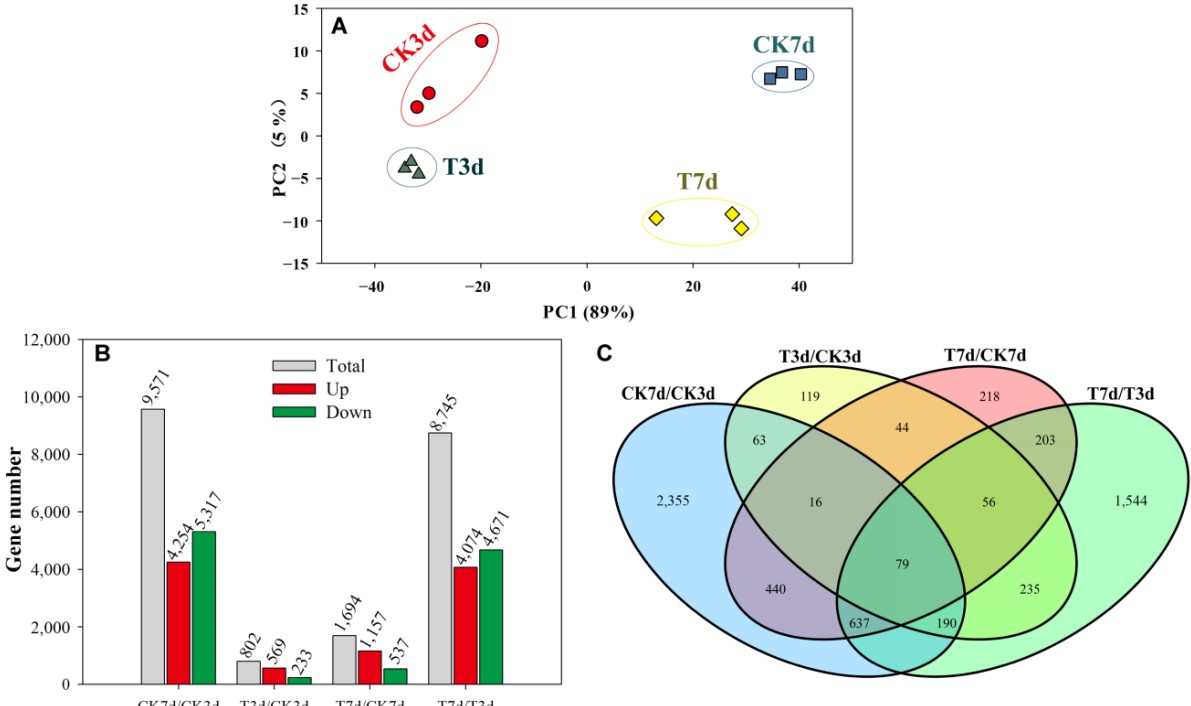

**Figure 5.** Principal component (PC) analysis and differentially expressed genes (DEGs) in the seedling leaves exposed to biuret stress. (**A**) The sample distribution in leaves according to PC1 and PC2. The percentage of variance is labeled for each component. (**B**) Numbers of DEGs in the seedlings under biuret treatment. (**C**) Venn diagrams for DEGS in the four comparison groups. CK3d, T3d, CK7d, and T7d represent the rice seedlings treated with 0 or 60 mg kg$^{-1}$ of biuret for 3 or 7 days, respectively.

GO enrichment analysis was performed with the DEGs from the four pairwise comparisons to identify the major functional terms (Figure 6A). These genes were mainly involved in biological processes, cellular components, and molecular functions. Among them, no common GO term was found between 3 days (T3d/CK3d) and 7 days (T7d/CK7d) (Figure 6A) due to the small number of common DEGs (Figure 5C). Noticeably, GO terms associated with the photosynthetic membrane, chloroplast, chloroplast nucleoid, chloroplast part, chloroplast stroma, chloroplast thylakoid membrane in the cellular component category, and response to hydrogen peroxide in the biological process were significantly enriched in the pairwise comparison of T7d/CK7d (*p*-value < 0.01; PDR < 0.05) (Figure 6A). These terms imply that photosynthesis-related genes were significantly downregulated and chloroplast development was likely inhibited when exposed to biuret application.

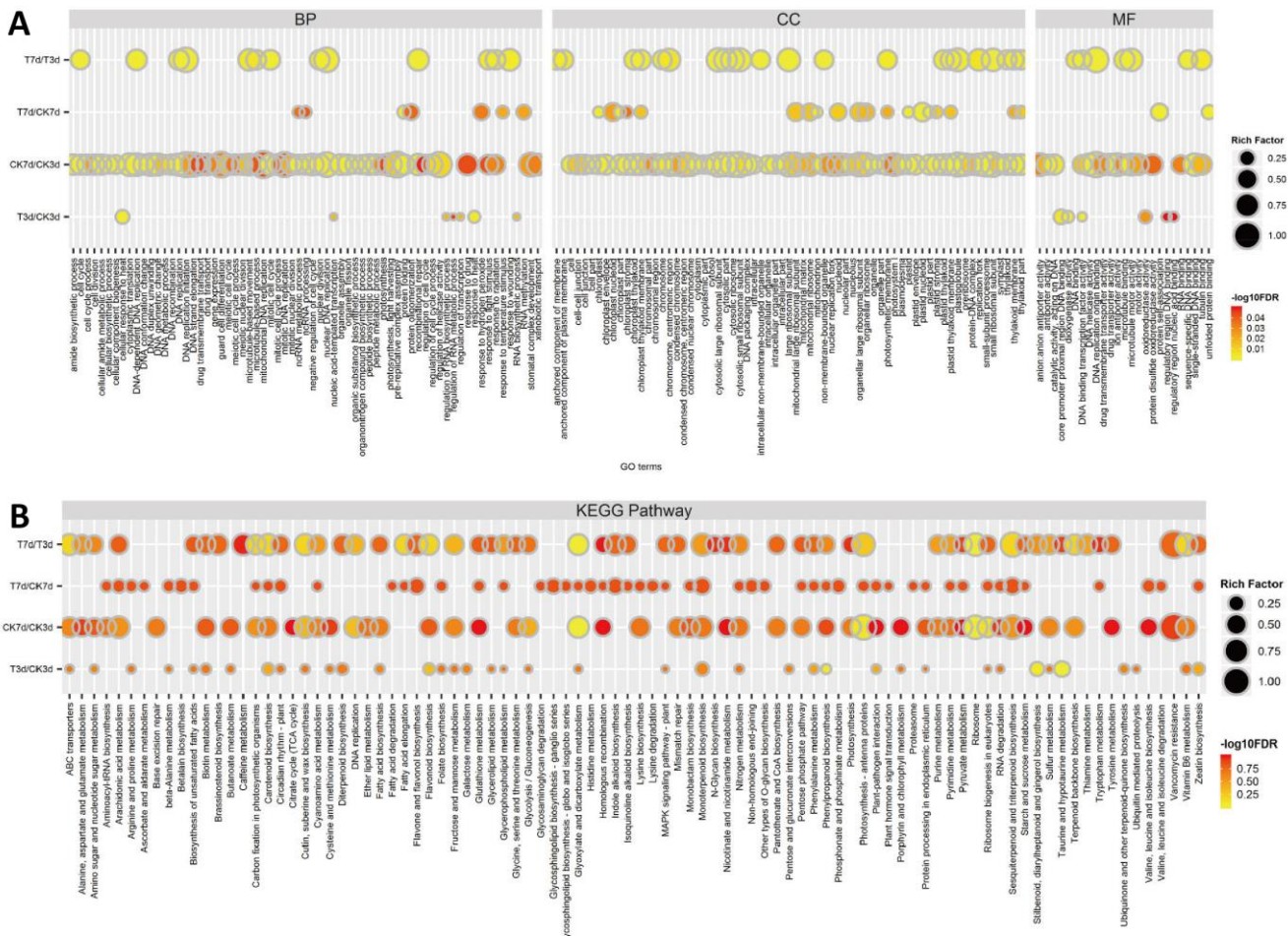

**Figure 6.** The enriched GO terms and KEGG enrichment analysis for the DEGs. (**A**) Important GO terms of DEGs with down-regulated expression in biuret treated seedlings compared with CK. BP, biological process; CC, cellular component; MF, molecular function. (**B**) Important KEGG pathways of DEGs with down-regulated expression in biuret treated seedlings compared with CK. CK3d, T3d, CK7d, and T7d represent the rice seedlings treated with 0 or 60 mg kg$^{-1}$ of biuret for 3 or 7 days, respectively.

For DEGs, the pathways involved in photosynthesis or pigments consisted of carbon fixation in photosynthetic organisms, carotenoid biosynthesis, flavonoid biosynthesis, photosynthesis, photosynthesis antenna proteins, porphyrin and chlorophyll metabolism, and pyruvate metabolism in at least two pairwise comparisons (Figure 6B). Furthermore, some key KEGGs were also found in nitrogen-related metabolism including the amide biosynthetic process, alanine, aspartate and glutamate metabolism, arginine and proline metabolism, N-glycan biosynthesis, and purine metabolism, indicating that biuret toxicity

may be related to ammonium transport or reactive nitrogen species (Figure 6B). Biuret-responsive DEGs were also enriched in glutathione metabolism, sulfur metabolism, and the MAPK signaling pathway(Figure 6B), which are important for stress adaptation in plants.

Taking into account the physiological data, GO terms (Figure 6A), and KEGG pathways (Figure 6B) of the transcriptome data, this study focused on the DEGs involved in chloroplast formation. A total of 34 identified DEGs were further divided into four main classes: 33 DEGs with significantly lower expression levels, namely, 13 chloroplast development, 11 photosystem, and 9 carbon fixation DEGs, and 2 DEGs related to chloroplast degradation with significantly higher expression levels in T7d than in CK7d (Figure 7A and Table S2).

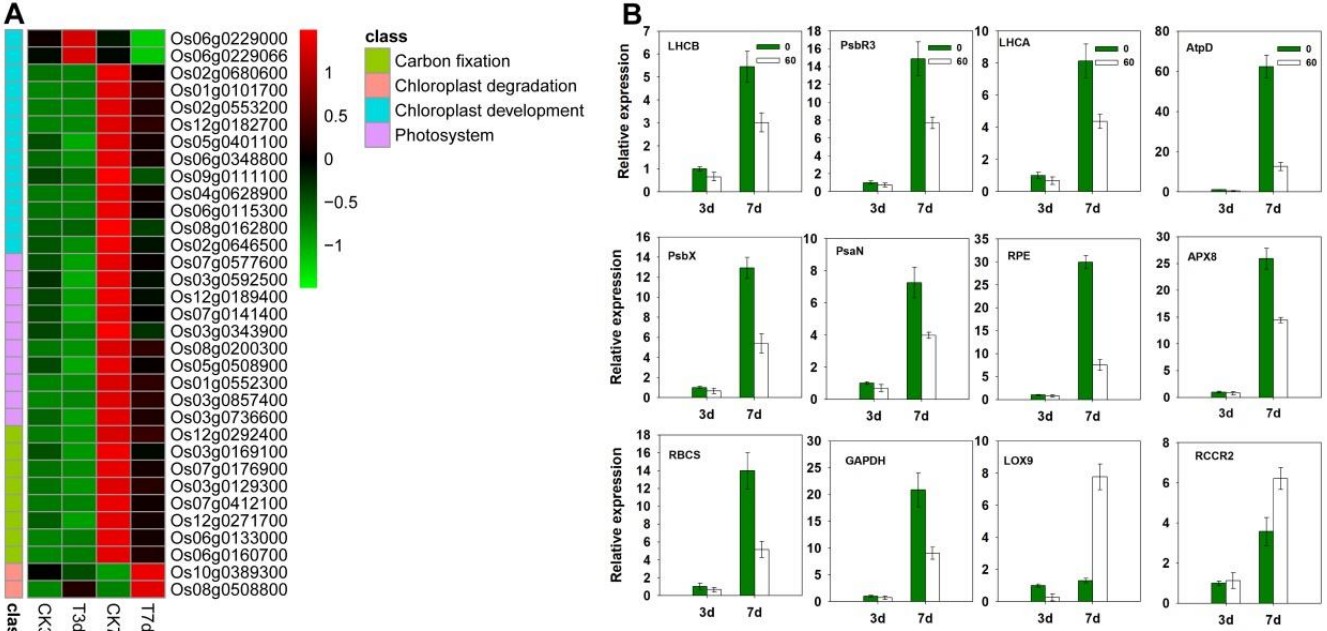

**Figure 7.** Heatmap of differentially co-expressed genes for different pathways (**A**) and quantitative real-time PCR validation (**B**). Relative expression values were calculated via Z-score normalization. Green and red represent low- and high-expression levels, respectively. The names of the samples are shown below. CK3d, T3d, CK7d, and T7d represent the rice seedlings treated with 0 or 60 mg kg$^{-1}$ of biuret for 3 or 7 days, respectively.

Twelve representative candidate genes were chosen from the DEGs in T3d/CK3d and T7d/CK7d for qRT-PCR analysis. These DEGs were mainly involved in the following metabolic pathways: carbon fixation in photosynthetic organisms, photosynthesis, photosynthesis antenna proteins, and chlorophyll metabolism. The qRT-PCR results of the selected genes obtained from the validation were all consistent with the RNA sequencing data (Figure 7B).

### 3.4. Biuret Decreased the Enzyme Activities of Chlorophyll Biosynthesis and Chloroplast Development

Rubisco activity was markedly stimulated when exposed to biuret on the 3rd day, and the activity of Rubisco was promoted by 23.8% and 43.7% with the increase in the biuret concentration, while the Rubisco activity was markedly decreased by 7.3% and 59.0% when exposed to biuret on the 7th day (Figure 8A). Correspondingly, the activities of GAPDH and RPE were significantly increased by 64.4% and 65.0% compared with the control on the 3rd day, while the activities of GAPDH and RPE were significantly lowered by 21.4% and 32.8% on the 7th day, respectively, when the rice seedlings were treated with 60 mg kg$^{-1}$ of biuret (Figure 8B,E). Compared with the control, the activities of RCCR and LOX in the leaves were significantly increased by 20.0% and 62.1%, and by 57.7% and 77.2%, on the 7th day when the rice seedlings were treated with 30 and 60 mg kg$^{-1}$ of biuret, respectively

(Figure 8C,D). These findings show that the APX activity in rice leaves was decreased by 17.7% with the application of 60 mg kg$^{-1}$ of biuret on the 7th day (Figure 8F).

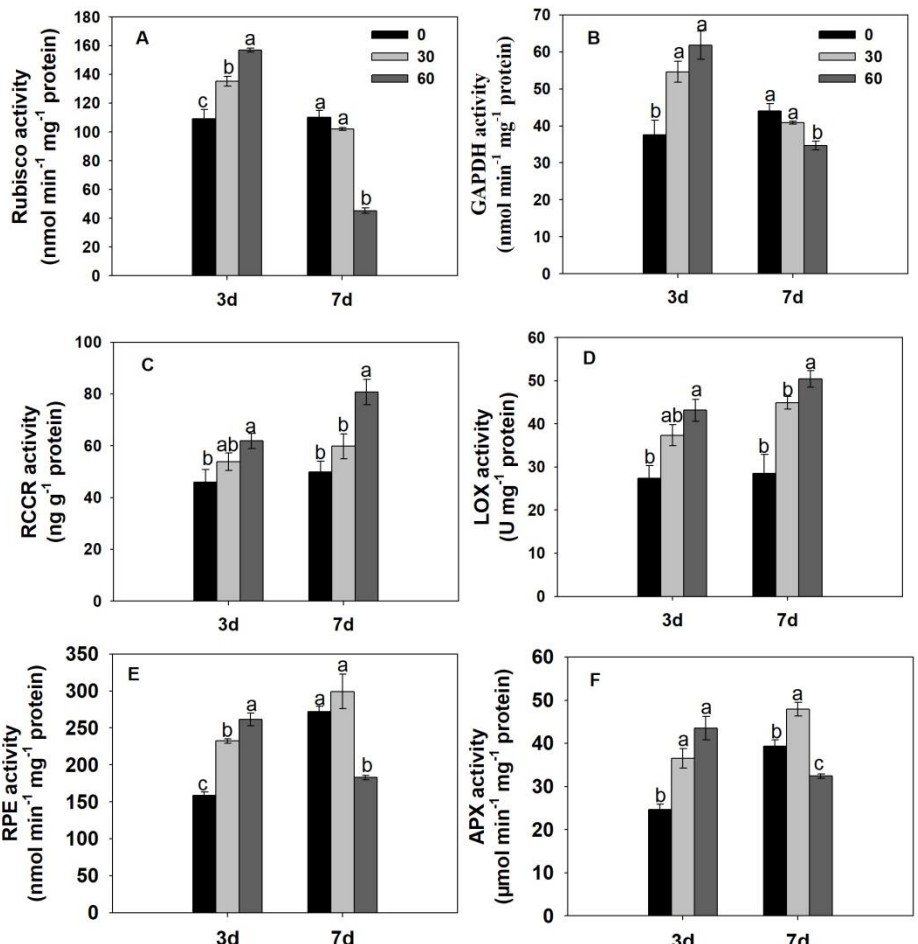

**Figure 8.** The enzyme activities in seedling leaves exposed to biuret stress. (**A**) Activities of Rubisco in seedling leaves under biuret stress. (**B**) Activities of GAPDH in seedling leaves under biuret stress. (**C**) Activities of RCCR in seedling leaves under biuret stress. (**D**) Activities of LOX in seedling leaves under biuret stress. (**E**) Activities of RPE in seedling leaves under biuret stress. (**F**) Activities of APX in seedling leaves under biuret stress. Values are means $\pm$ SE (three independent biological replicates). Ten seedlings were combined for a representative sample. Different letters above the bars designate significant differences at $p < 0.05$.

## 4. Discussion

Rice leaves are sensitive to biuret, and chlorosis in the leaves is induced by the application of fertilizers containing more than 2% biuret [7]. In this study, the chlorotic leaves of rice seedlings showed higher biuret accumulation (Figure 1). Abnormal chloroplast development and the inhibition of chlorophyll synthesis under stress can cause leaf color mutation in plants [10,30]. The results of this study indicate that the albino leaves exhibited abnormal cells with serious swelling of the chloroplasts and the disappearance of internal structure (Figure 2), and the chlorophyll contents were significantly decreased due to excessive biuret. The chloroplasts in albino rice leaves may have been affected by biuret at the basal grain construction stage, which inhibited the development of chloroplasts.

Impaired development and maturation of chloroplasts in plant leaves can lead to abnormal leaf color [31,32]. Studies have shown that the *GLK* gene family acts as a regulating factor for chloroplast development in rice [33], maize [34,35], and *Arabidopsis thaliana* [36]. The leaves of maize g2 (*golden2* and *Zmglk2*) mutants have smaller chloroplasts compared with the wild type and only develop basic thylakoid lamellae with very few grana [37].

The transcriptome analysis in our study found lower GLK (*OsGLK1*) gene expression in the rice exposed to biuret stress compared with the control, which was consistent with the qPCR results. The thylakoid membrane is the site of photosynthetic electron transfer and photosynthetic phosphorylation in photosynthesis, and the stability of its membrane structure has an important impact on photosynthesis. Aldehyde dehydrogenases (ALDHs) oxidize aldehydes into carboxylic acids, which supply the substrate for acetyl-CoA synthetase (ACS) to synthesize acetyl-CoA [38]. The acyl-CoA-binding proteins (ACBPs) participate in the synthesis of lipid signals and are thus important in regulating fatty acid biosynthesis [39]. Under biuret stress, the expression of ALDH and ACBP genes in rice leaves was strongly inhibited, and the supply of precursors for fatty acid biosynthesis was insufficient. Rice leaves may accumulate large amounts of ROS, including hydrogen peroxide, after exposure to biuret. The transcript level of the thylakoid-membrane-bound ascorbate peroxidase gene (*APX8*) was downregulated, and the expression of lipoxygenase genes (*Lox9*) was significantly increased during biuret treatment, which indicated an increased level of hydrogen peroxide and lipid peroxidation. Lipid synthesis decreased, and the development of chloroplast membranes and thylakoid membranes in chloroplasts was abnormal, which affected the function of the membrane proteins.

The photosynthetic chlorophyll protein complexes (photosystem I and photosystem II (PSII)) are located in the thylakoid membranes of chloroplasts. PSII consists of three large protein complexes, namely, the core complex, the PSII light-harvesting complex (LHCII), and the oxygen-evolving complex (OEC). It mainly produces strong oxidants after illumination and catalyzes the oxidation of water and the reduction of plastoquinone [40]. The chlorophyll a/b-binding protein is encoded by the nuclear genome and acts as an apoprotein of the light-harvesting chlorophyll complex (Lhc), which largely prevents protease degradation in the thylakoid membrane [41,42]. Chlorophyll content is positively correlated with Lhcb gene expression levels [43]. The decrease in chlorophyll content caused by biuret treatment could suppress the expression of the Lhcb gene. The transcriptome analysis in this study found that the expression of two genes, annotated as the light-harvesting chlorophyll a/b-binding protein (Lhcb), was downregulated with biuret application. The OEC of PSII is composed of four extrinsic nuclear-encoded subunits, namely, PsbO (33 kDa), PsbP (23 kDa), PsbQ (17 kDa), and PsbR (10 kDa). PsbR is an important link for the psbP protein to stably assemble in the formation of the OEC [44]. The psbR and psbQ proteins depend on each other, and a lack of the psbR protein has a significant impact on the activity of PSII [45]. The expression of the psbR gene was greatly downregulated, and the expression of the *PsbP* and *PsbX* genes was also significantly inhibited in the biuret-affected leaf tissues. The maximum ($F_v/F_m$) and effective quantum yield of PSII ($\Phi_{PSII}$) indicated a significant decline under biuret application, indicating that the formation of the OEC in the chloroplast PSII was hindered. The expression of nuclear genes encoding the PSI reaction center N subunit (psaN) was downregulated under biuret application. PsaN is located near the cavity side of PsaF, where it participates in the docking of plastocyanin [46]. Similarly, the leaves were also chlorotic, and the photosynthetic rate was significantly reduced when the psaN gene was absent in Arabidopsis in [47,48]. Thus, the stability of the chloroplast PSII oxygen-evolving complex and the PSI complex was affected, resulting in abnormal chloroplast structures and leaf whitening.

The Calvin cycle is part of the dark reactions of photosynthesis and the cyclic process of $CO_2$ assimilation in higher plants. Rubisco is a soluble chloroplast enzyme that catalyzes the first step in net photosynthetic $CO_2$ assimilation [49]. GAPDH is also a key enzyme in glycolysis and gluconeogenesis during the Calvin cycle [50]. Calvin-cycle-dependent genes (*GAPDH*, *RPI*, and *RPE*) were greatly reduced under biuret treatment. In particular, the expressions of ribulose bisphosphate carboxylase small chain (*RbcS*) and GBSSII, RPE, which encode the small subunit of Rubisco, and GBSSII, RPE as a chloroplast precursor, were reduced with biuret application (Figure 8). These genes related to photosynthesis and carbon fixation were crucial for rice growth, and the inhibition of their expression might explain the reduction in carbon sources.

## 5. Conclusions

Some biuret stress responses in leaf DEGs were assessed with comparative morphophysiological and transcriptome analysis. The abnormal chloroplast ultrastructures and the loss of the thylakoid membrane structure in rice leaves promoted the appearance of an albino phenotype under exposure to biuret. The DEGs involved in the chloroplast development, photosynthesis (including antenna proteins), and carbon fixation pathways in photosynthetic organisms were all significantly downregulated, which suggested that photosynthesis was destroyed in the chlorotic leaves of rice seedlings under biuret treatment. A large number of genes related to chloroplast development and photosynthesis were regulated under biuret stress, indicating that the photosynthetic system in the chloroplast thylakoid membrane was disturbed in chlorotic rice leaves. These results show that the inhibition of chloroplast development induced the disorder of the photosynthetic system under biuret stress, which was the key reason for the formation of the chlorotic leaf phenotype in rice.

**Supplementary Materials:** The following supporting information can be downloaded at https://www.mdpi.com/article/10.3390/agronomy13082052/s1. Table S1: Summary of transcriptome sequencing results; Table S2: Heatmap data of differentially co-expressed genes for different pathways.

**Author Contributions:** Y.Z. (Yikai Zhang) and H.C. conceived the project and designed the experiments; H.C., Y.Z. (Yuping Zhang) and Y.Z. (Yikai Zhang) provided funding; Y.Z. (Yikai Zhang), Y.C. and J.X. performed the experiments; Z.W. and Y.Z. (Yuping Zhang) analyzed the RNA-seq data; H.C. and Y.W. contributed reagents/materials/analysis tools; P.Z. drew the figures and wrote the manuscript. All authors have read and agreed to the published version of the manuscript.

**Funding:** This work was supported by the National Key Research and Development Plan of China (2022YFD1500404), the Research and Development Project of Zhejiang Province (2022C02008; 2023C02004; 2022C02034), the Agricultural Major Technology Collaborative Promotion Plan of Zhejiang Province (2021XTTGLY0103), Special Funds for the Construction of Modern Agricultural Technology Systems (CARS-01-22), and the Agricultural Sciences and Technologies Innovation Program of the Chinese Academy of Agricultural Sciences (CAAS) for the Rice Intelligence and High-Efficiency Cultivation Technology Group.

**Data Availability Statement:** The data recorded in the current study are available in all the figures and Supplementary Tables in the manuscript.

**Conflicts of Interest:** The authors declare no conflict of interest.

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
