# Peer review of "Chloroplast Damage and Photosynthetic System Disorder Induced Chlorosis in the Leaves of Rice Seedlings under Excessive Biuret"

_agronomy, doi:10.3390/agronomy13082052_

Round 1

Reviewer 1 Report

Potenatialy interesting paper, but require completely re-writing before acceptance.

 Many sentences require edition of English.

Line 10: „The aim of this study was to provide a comprehensive understanding underlying the physiology and molecular mechanisms of leaf chlorosis by biuret using morphophysiological and transcriptome analyses” = “This study aimed to provide a comprehensive understanding of the physiological and molecular mechanisms of leaf chlorosis by biuret using morphophysiological and transcriptome analyses.”

Line 13: “The effects of biuret on albino rice leaves” do you mean induction of albinism by biuret?

Line 21: “The related genes were significantly down-regulated involved in the chloroplast development, photosynthesis (including antenna proteins), and carbon fixation pathways in photosynthetic organisms, which suggested that photosynthesis was destroyed in the chlorotic leaves of rice seedlings.” – please, edit grammar = “The related genes involved in the chloroplast development, photosynthesis (including antenna proteins), and carbon fixation pathways were significantly down-regulated, which suggested that photosynthesis was destroyed in the chlorotic leaves of rice seedlings.”

Line 25: “ must be “Biuret disturbed the photosynthetic system in the chloroplast thylakoid membrane by inhibiting chloroplast development, thereby promoting the formation of the chlorotic leaf phenotype in rice seedlings.”.

Line 40: I suggest to edit like this to avoid repetition: “In compound fertilizer production, urea has become the preferred nitrogen source due to its high nitrogen content”.

Line 44: point require after sentence end.

Line 61: “In addition to the influence of external environmental factors such as light, temperature, moisture, and mineral elements, plant leaf albinism is mainly controlled by internal genes.” ?? Please, edit!

Line 83: “1.64 g kg−1 total N, 18.92 mg kg−1 available P, 50.02 mg kg−1 exchangeable K” – it seems to be far from optimal, very low P and low K. How can you explain this?

is better to write how many seeds per bed, not per ha.

Line 89: “During the experiment, biuret was added at rates of 0, 30, and 60 mg kg−1, and then

the biuret was sprinkled onto the cultivation bed and tilled evenly.” Per kg of soil? Please, provide more details.

Line 93: “the physiological index” ? This part is not clear.

Lines 98- 99: “ At harvest, seedlings were separated into shoot and root, which were rinsed with

deionized water.” Please, clarify.

Line 102: “Then, the centrifuge was used to rotate 8000 rpm for 10 min, and the supernatant was taken in an evaporating dish and dried in a water bath.” = “Then, the samples were centrifugated at 8000 rpm for 10 min, and the supernatant was taken in an evaporating dish and dried in a water bath.”. At which temperature?

Line 183: “Glyceraldehyde 3-phosphate dehydrogenase (GAPDH)”

Line 184- 190: the citation is enough for the method, it is not necessary to always repeat “previously described procedure”.

Lines 263- 282: how authors can separate primary and secondary genes after biuret treatment? Primary mean directly affected by biuret and secondary mean results of whole metabolism changes?

How biuret treatment affected carbon deficiency and how external applications of caron as carbohydrate may minimize biuret effect?

Lines 327 - 338: how did you count activity? It looks after 7 days biuret exposure definitely exist death cell in plants what may lead to wrong estimation. Which APX do yu mean? There are cytoplasmic, stroma and thylakoid APX. Please, clarify.

Line 354: “Impaired chloroplast differentiation and development in plant leaves” ? Maybe development and maturation?

Line 359 this = our.

re-writing are required

Author Response

Dear Editor:

We are very grateful to the reviewers’ comments for the manuscript. According to your suggestions, we have revised the manuscript. Below are the responses to each question: 

Reviewer #1:

Q1: Line 10: “The aim of this study was to provide a comprehensive understanding underlying the physiology and molecular mechanisms of leaf chlorosis by biuret using morphophysiological and transcriptome analyses” = “This study aimed to provide a comprehensive understanding of the physiological and molecular mechanisms of leaf chlorosis by biuret using morphophysiological and transcriptome analyses.”

A: Thanks for your suggestion! We've revised it to “This study aimed to provide a comprehensive understanding underlying the physiology and molecular mechanisms of leaf chlorosis by biuret using morphophysiological and transcriptome analyses” in Line 12-13.

Q2: Line 13: “The effects of biuret on albino rice leaves” do you mean induction of albinism by biuret?

A: Indeed, this sentence was intended to clarify the phenomenon of biuret-induced albino rice leaves. We realize that this sentence may not have been phrased accurately enough and have revised it to "The induction of biuret on albino rice leaves was examined in a net-growing cultivation bed." in Line 14.

Q3: Line 21: “The related genes were significantly down-regulated involved in the chloroplast development, photosynthesis (including antenna proteins), and carbon fixation pathways in photosynthetic organisms, which suggested that photosynthesis was destroyed in the chlorotic leaves of rice seedlings.” – please, edit grammar = “The related genes involved in the chloroplast development, photosynthesis (including antenna proteins), and carbon fixation pathways were significantly down-regulated, which suggested that photosynthesis was destroyed in the chlorotic leaves of rice seedlings.”

A: Thanks for your suggestion!  We have checked the grammar of this sentence and revised it to “The related genes involved in the chloroplast development, photosynthesis (including antenna proteins), and carbon fixation pathways were significantly down-regulated, which suggested that photosynthesis was destroyed in the chlorotic leaves of rice seedlings.” in Line 24.

Q4: Line 25: must be “Biuret disturbed the photosynthetic system in the chloroplast thylakoid membrane by inhibiting chloroplast development, thereby promoting the formation of the chlorotic leaf phenotype in rice seedlings.”.

A: Thank you! We have realized the wrong use of the preposition “through” and have revised “through” to “by” in Line 27.

Q5: Line 40: I suggest to edit like this to avoid repetition: “In compound fertilizer production, urea has become the preferred nitrogen source due to its high nitrogen content”.

A: Thanks for the suggestion! We have revised the sentence as “In compound fertilizer production, urea has become the preferred nitrogen source due to its high nitrogen content.” in Line 42.

Q6: Line 44: point require after sentence end.

A: Thank you! We've added “punctuation mark” in Line 46.

Q7: Line 61: “In addition to the influence of external environmental factors such as light, temperature, moisture, and mineral elements, plant leaf albinism is mainly controlled by internal genes.” ?? Please, edit!

A: Thank you! We've revised this sentence to “External environmental influences such as light, temperature, water and mineral elements account for albinism in plant leaves, however, apart from these external factors, leaf albinism in plants is inherently and primarily controlled by internal genes.” in Line 63.

Q8: Line 83: “1.64 g kg−1 total N, 18.92 mg kg−1 available P, 50.02 mg kg−1 exchangeable K” – it seems to be far from optimal, very low P and low K. How can you explain this? It is better to write how many seeds per bed, not per ha.

A: We are sorry that the physical and chemical properties of basic soil are not clear and thank you for good advice. The available P was Olsen-P, which was extracted with 0.5 mol•L-1sodium bicarbonate and determined by anti-colorimetric analysis of molybdenum and antimony at 880 nm. Critical levels for Olsen-P reported for rice range from 5 mg P kg-1 in acid soils. The exchangeable K was available potassium, which was extracted with 1.0 mol•L-1 ammonium acetate and determined by flame photometry (Model 410 Flame Photometer). Critical levels for available K reported for rice range from 50 mg K kg-1. The available P content of the soil used in the experiment is moderate, and there is a lack of potassium. We applied potassium fertilizer (Potassium chloride) 120 mg/kg P and phosphorus fertilizer (Superphosphate) 150 mg/kg K before rice planting. Therefore, the addition of these nutrients, P and K, can meet and ensure the optimal growth of rice seedlings. We have revised the description of seeding level to “The seeds were soaked for 48 h and directly seeded in the cultivation bed at a planting density of 80 g per bed.” in Line 88.

Q9: Line 89: “During the experiment, biuret was added at rates of 0, 30, and 60 mg kg−1, and then the biuret was sprinkled onto the cultivation bed and tilled evenly.” Per kg of soil? Please, provide more details.

A: Thank you! We have revised it to "at the rates of 0, 30, and 60 mg biuret kg-1 of soil.” and provide the details “During the experiment, biuret was added at rates of 0, 30, and 60 mg biuret kg-1 of soil. The amount of soil nutrients added was as follows (mg kg−1 soil): 200 N (as (NH4)2SO4), 120 P (as superphosphate), and 150 K (as KCl). Initially, the biuret required for the experiment was weighed, and subsequently blended evenly with the fertilizer. Prior to sowing, the seedbed was irrigated, plowed, and leveled, and then the fertilizer mixture was uniformly distributed onto the seedbed and incorporated into the soil to a depth of 10 cm, and sowed.” in Line 91-97.

Q10: Line 93: “the physiological index”? This part is not clear.

A: Thank you! We realized that further description of physiological indexes may be required to be provided here. We have revised this sentence to “Subsequently, seedlings were sampled and physiological indexes such as plant height, root length, and shoot weight were determined. Gene expression was also analyzed using high-throughput sequencing.” in Line 100-102.

Q11: Lines 98- 99: “At harvest, seedlings were separated into shoot and root, which were rinsed with deionized water.” Please, clarify.

A: Thank you! We realized that "At harvest" may lead to misinterpretation by readers and have replaced "At harvest" with "During sampling" in Line 104.

Q12: Line 102: “Then, the centrifuge was used to rotate 8000 rpm for 10 min, and the supernatant was taken in an evaporating dish and dried in a water bath.” = “Then, the samples were centrifugated at 8000 rpm for 10 min, and the supernatant was taken in an evaporating dish and dried in a water bath.”. At which temperature?

A: Thank you! We have added the temperature and revised the sentence to “Then, the samples were centrifugated at 8000 rpm for 10 min with a temperature of 25 °C, and the supernatant was taken in an evaporating dish and dried in a water bath.” in Line 108.

Q13: Line 184- 190: the citation is enough for the method, it is not necessary to always repeat “previously described procedure”.

A: Thank you! We realized that the repetitive words have caused this sentence to be less readable, taking your suggestions into account and streamlining them as follows: “Activities of D-ribulose-5-phosphate 3-epimerase (RPE), lipoxygenase activity (LOX), red chlorophyll catabolite reductase (RCCR) and ascorbate peroxidase (APX) were evaluated according to the previous studies [25-27].”. (Line 191-194)

Q14: Lines 263- 282: how authors can separate primary and secondary genes after biuret treatment? Primary mean directly affected by biuret and secondary mean results of whole metabolism changes? How biuret treatment affected carbon deficiency and how external applications of caron as carbohydrate may minimize biuret effect?

A: Thank you for reminding me. We are very sorry for our mistakes. Figure 5 was lost due to our carelessness, which has been reinserted as follows. We performed Principal Component Analysis (PCA) using transcriptome data to provide an overview of the transcriptomic landscape. As shown in Figure 5a, samples from different germination days were separated by the first principal component (PC1). PC2 explained a relatively small proportion of the overall variance found in the data set (5%). However, the samples taken 7 days after germination became more scattered than those taken at 3 days, indicating that biuret had a greater effect on samples taken 7 days after germination. Moreover, the number of differential expression gene among the different treatments was also confirmed based on PCA (Figure 5b). Therefore, we analyzed the effects of different treatments, not primary and secondary genes. In 2.5 section, we also added the methods of PCA “The principal components analysis (PCA) was performed to explicit the variation of transcriptome”.

Figure 5. Principal component (PC) analysis and differentially expressed genes (DEGs) in the seedling leaves exposed to biuret stress. (A) The sample distribution in leaves according to PC1 and PC2. The percentage of variance is labeled for each component. (B) Numbers of DEGs in the seedling under biuret treatment. (C) Venn diagrams for DEGS in the four comparison groups. CK3d, T3d, CK7d and T7d represent the rice seedlings treated with 0, 60 mg kg−1 biuret for 3 or 7 days, respectively.

Q15: Lines 327 - 338: how did you count activity? It looks after 7 days biuret exposure definitely exist death cell in plants what may lead to wrong estimation. Which APX do yu mean? There are cytoplasmic, stroma and thylakoid APX. Please, clarify.

A: The measurement method we use was the total enzyme activity of APX (ascorbate peroxidase), which was measured at the rate of ascorbate oxidation at 290 nm. The assay mixture contained 0.25 mM AsA, 1.0 mM H2O2, 0.1 mM EDTA, and 0.1 ml enzyme extract in 25 mM phosphate buffer (pH 7.0). The activity of APX was calculated in terms of μmol min-1 mg-1 protein. Under abiotic stress, the activity of APX enzyme undergoes a stress response after being subjected to stress for a short period of time. With the extension of time, the enzyme activity decreased, while reactive oxygen species and membrane peroxidation products increased (Figue 1). In high-throughput sequencing of Transcriptome, the expression trend of APX related genes is similar to that of the enzyme activity.

Q16: Line 354: “Impaired chloroplast differentiation and development in plant leaves”? Maybe development and maturation?

A: Indeed, development and maturation are more accurate. We have revised it to “Impaired development and maturation of chloroplast in plant leaves can lead to ab-normal leaf color” in Line 380.

Q17: Line 359 this = our.

A: Thanks for the heads up, we have replaced "in this study" with "in our study" in Line 385.

We hope that we have revised the manuscript to the reviewer’s request. Should you need any further information and clarification, please do not hesitate to contact us. Many thanks for your time and consideration!

                               Best wishes,

Yikai Zhang

State key laboratory of rice biology and breeding, China National Rice Research Institute, Hangzhou, Zhejiang 310006, China, E-mail: zhangyikai@caas.cn, Tel: +86 571 6337 0012

Reviewer 2 Report

This manuscript is a very successful study in every respect!

Note a few remarks for corrections in the file.

Author Response

Dear Editor:

We are very grateful to the reviewers’ comments for the manuscript. According to your suggestions, we have revised the manuscript. Below are the responses to each question: 

Reviewer #2:

Q1: Line89-“0, 30, and 60 mg kg−1”, of what? The unit needs further definition.

A: Thank you! We have revised it to "at the rates of 0, 30, and 60 mg biuret kg-1 of soil" in Line 91.
Q2:Line90- “the biuret was sprinkled onto the cultivation bed and tilled evenly”, sprinkled?

A: Ok! We have refined this description and revised it as “During the experiment, biuret was added at the rates of 0, 30, and 60 mg biuret kg-1 of soil. The amount of soil nutrients added was as follows (mg kg−1 soil): 200 N (as (NH4)2SO4), 120 P (as superphosphate), and 150 K (as KCl). Initially, the biuret required for the experiment was weighed, and subsequently blended evenly with the fertilizer. Prior to sowing, the seedbed was irrigated, plowed, and leveled, and then the fertilizer mixture was uniformly distributed onto the seedbed and incorporated into the soil to a depth of 10 cm, and sowed.” in Line 94-99.

Q3: Line94- “the biuret treatment was harvested at 3 days and 7 days after seedling emergence”, The use of the words “were harvested at 3 days and 7 days” and “seedling emergence” may raise some barriers to the reader's understanding.

A: We have revised it to “the biuret treatments were sampled at 3 and 7 days after seedling emergence, respectively” in Line 99.

Q4:Line98- It might be better to use “sampling” instead of “harvest”.

A: We have revised the sentence of “At harvest” to "During sampling" in Line 99.

Q5:Line111- There should be “was” instead of “is”.

A: Thank you! We have revised it to “was” in Line 118.

Q6:Line216- There should be “plant height” instead of “plant heights”. Line217- It might be better to use “rice seedlings” instead of “rice plants”.

A: Thank you! We have revised them to “plant height” and “rice seedlings” in Line 224.

Q7:Line252- “80 and 160 mg kg−1 biuret”, This data of 80 and 160 does not appear in Figure 4, so please check if you have entered the false data.

A: We are very sorry for the mistakes, and have revised them to “30 and 60 mg kg−1 biuret” in Line 262.

Q8:Line356- “Arabidopsis thaliana”, should go in italics.

A: Thank you! We've revised the manuscript with reference to your suggestion in Line 382.

We hope that we have revised the manuscript to the reviewer’s request. Should you need any further information and clarification, please do not hesitate to contact us. Many thanks for your time and consideration!

                               Best wishes,

Yikai Zhang

State key laboratory of rice biology and breeding, China National Rice Research Institute, Hangzhou, Zhejiang 310006, China, E-mail: zhangyikai@caas.cn, Tel: +86 571 6337 0012

Reviewer 3 Report

The paper is interesting as it focus in an important problem in a major crop. The authors relate the biuret toxicity, typical of low quality fertilizers, with the leaf chlorosis. The main criticism of the paper is that authors only evaluate their transcriptomic results in relation with the choloroplast, but, could exist any other physiological process differentially regulated due to the biuret toxicity. For instance Amonium transport proteins? reactive nitrogen species? Paper demands to make a deeper analysis of which GO categories are enriched, besides the ones related to chloroplast.

Minor points:

-Fig 1B. Shoot weight. the gray column in 7 days should be an "a" instead of a "b".

- Authors should include the number of plants per data point (n) in each figure legend.

- line 356: Arabidopsis thaliana, should go in italics.

- line 405: rubisco is not a nuclear encoded protein. Small chains are encoded in the nucleus, but large chains are encoded in the chloroplast. Please correct. 
